# Impact of COVID-19 Lockdown on Food Consumption and Behavior in France (COVISTRESS Study)

**DOI:** 10.3390/nu14183739

**Published:** 2022-09-10

**Authors:** Mélanie Pouget, Maëlys Clinchamps, Céline Lambert, Bruno Pereira, Nicolas Farigon, Elodie Gentes, Magalie Miolanne, Mathilde Picard, Anne Tyrode, Maud Alligier, Frédéric Dutheil, Yves Boirie

**Affiliations:** 1Service de Nutrition Clinique, CSO CALORIS, CHU Clermont-Ferrand, Université Clermont Auvergne, 63000 Clermont-Ferrand, France; 2Service Santé Travail Environnement, CHU Clermont-Ferrand, Université Clermont Auvergne, 63000 Clermont-Ferrand, France; 3Unité de Biostatistiques, DRCI, CHU Clermont-Ferrand, 63000 Clermont-Ferrand, France; 4AIST La Prévention Active, 63000 Clermont-Ferrand, France; 5FCRIN/FORCE Network, Centre de Recherche en Nutrition Humaine Rhône-Alpes, 69000 Lyon, France; 6Service Santé Travail Environnement, CHU Clermont–Ferrand, LaPSCo, CNRS, Université Clermont Auvergne, WittyFit, 63000 Clermont–Ferrand, France; 7Unité de Nutrition Humaine, UMR1019, CRNH Auvergne, INRA, Université Clermont Auvergne, 63000 Clermont-Ferrand, France

**Keywords:** COVID-19, lockdown, eating habits

## Abstract

The COVID-19 pandemic and subsequent lockdowns modified work environments, lifestyles, and food consumption. Eating habits and mood changes in a French population during the first lockdown were examined using an online self-reported questionnaire with REDCap software through the COVISTRESS.ORG website. In 671 French participants, the main changes during lockdown were increased stress levels (64 [23; 86] vs. 3 [0; 18]) and sedentary behavior (7 [4; 9] vs. 5 [3; 8] hours per day), a deterioration in sleep quality (50 [27; 83] vs. 70 [48; 94]) and mood (50 [30; 76] vs. 78 [50; 92]), and less physical activity (2.0 [0.5; 5.0] vs. 3.5 [2.0; 6.0]). Mood was modified, with more anger (56 [39; 76] vs. 31 [16; 50]), more sadness (50 [34; 72] vs. 28 [16; 50]), more agitation (50 [25; 66] vs. 43 [20; 50]), and more boredom (32 [7; 60] vs. 14 [3; 29]). A total of 25% of the participants increased their consumption of alcoholic beverages, 29% their consumption of sugary foods, and 26% their consumption of cocktail snacks. A multiple-correspondence analysis highlights four different profiles according to changes in eating habits, food consumption, lifestyle, and mood. In conclusion, eating habits and lifestyle changes during lockdown periods should be carefully monitored to promote healthy behaviors.

## 1. Introduction

The 2019 coronavirus disease pandemic (COVID-19), caused by severe acute respiratory syndrome coronavirus 2 (SARSCoV-2), started in late 2019 in Wuhan (capital of Hubei, China) [1,2]. According to a World Health Organization report from 9 June 2022, 530,896,347 cases have been confirmed since the outbreak, more than 6,301,020 of which were fatal. Lockdown was enforced in more than half of the world’s countries to slow the exponential spread of the COVID-19 virus and has been a major public health measure to help health systems manage hospitalized patients. In France, two lockdowns were ordered: the first between 17 March and 10 May 2020, and the second between 30 October and 15 December 2020. During the first lockdown, exceptional measures were taken: limitation of movement (except for travel to work or attending medical appointments), closure of non-essential businesses and venues open to the public such as bars and restaurants or sports facilities, and working from home encouraged whenever possible. However, during the second lockdown, the rules were less strict for travel outside the home than during the first lockdown: Schools remained open with reinforced health protocols. Lockdown reduced movement outside the home, with repercussions for the daily and social life of French people, leading to changes in eating habits (timing of meals, mode of supply, reduced access to fresh products) and reduced access to regular physical activity.

Quarantine is associated overall with stress and depression, leading to an unhealthy diet and reduced physical activity. A diet poor in fruit and vegetables is frequently reported during a period of isolation like a lockdown, with a consequent low intake of antioxidants and vitamins [3]. Stress and anxiety induce people to eat sugar-rich or high-fat food to feel better. A commanding desire to consume a specific kind of food is termed a “food craving.” It has several dimensions: cognitive (thinking about food), emotional (desire to eat or changes in mood), behavioral (seeking and consuming food), and physiological (salivation) [3].

The “CoviPrev” survey was started to monitor changes in behavior (sanitary precautions, lockdown, alcohol and tobacco consumption, diet, and physical activity) and mental health (well-being, disorders). The main changes reported relate to snacking, home cooking, accessibility of food products, and weight change [4]. According to the results collected in the CoviPrev survey, 17% of people questioned considered their diet to be less balanced than before lockdown (against 13% declaring a more balanced diet than usual), 22% snacked between meals more than usual (against 17% less than usual), 37% said they cooked home meals more than usual (against 4% less than usual), and 27% said they had gained weight (against 11% who had lost weight).

A recent study of 938 French adults showed that the nutritional quality of their diet was lower during the lockdown than before [5]. Food-choice motives significantly changed: Increased importance of weight control was associated with better nutritional quality, whereas increased importance of mood was associated with poorer nutritional quality. The participants increased their intake of fruit and vegetables, pulses, fish, and seafood, or they increased their consumption of processed meat, sugary foods, sweet-tasting beverages, and alcoholic beverages, leading to a decrease in the nutritional quality of their diet. In the French cohort NutriNet-Santé, during the first lockdown, unfavorable changes were also reported [6].

The aim of this study was to examine the eating habits and mood changes of the French population during the first lockdown. A second aim was to determine specific profiles of patients who suffered most from lockdown.

## 2. Method

### 2.1. Participants

All participants were volunteers and were informed of the objective of the survey and that their data would be processed anonymously and be used for research purposes only. Participants could withdraw from the study at any time without having to give reasons. Participants included in the analysis were French adults who answered questions about their weight before and during the first lockdown and who answered all questions about their consumption in the different food categories before and during the first lockdown.

The study was reviewed and approved by the human ethics committees Sud-Est VI, France (2020/CE 11-clinicaltrials.gov NCT 04308187). Selection of participants, recruitment, and advertising were carried out through radio, television, fliers, posters, newspapers, and online media, including social media such as Facebook, Twitter, etc. The ethics committee waived the need for written consent, considering that if people responded to the questionnaires by going to the website, then they were consenting. 

### 2.2. Study Design and Setting

We implemented an open epidemiological, observational, descriptive study by administering a self-reported questionnaire proposed to volunteers using the REDCap software available through the COVISTRESS.ORG website. The REDCap questionnaire was hosted by the University Hospital of Clermont-Ferrand. The questions analyzed in this work were therefore specific questions included in a large questionnaire with various thematic sections. The thematic sections were presented in random order after the demographic questions. The online questionnaire was distributed several times through mailing lists held by institutions and French social groups. The data analyzed were obtained between 26 June 2020 and 2 March 2021.

### 2.3. Variables

The primary outcomes measured were diet habits, lifestyle, and mood. Dietary habits were ascertained by asking the number of times per week that different food categories were consumed before the pandemic and during the first lockdown. Cocktail snacks include crisps, savory biscuits, and peanuts. For food consumption (number of times per week the food was eaten), a change (increase or decrease) was defined when the number was not the same before and during the first lockdown.

The participants gave general information: age, sex, weight, height, employment status, level of education, type of housing, household composition, and consumption of alcohol, cigarettes, and cannabis. 

Participants were asked various questions about nutrition: (i) How do you describe your diet? (visual analogue scales (VAS) from 0 “not at all balanced” to 100 “very balanced”); (ii) For you, eating is: Essential for living? Useful for maintaining health? A pleasure? A moment of sharing? A constraint? (VAS from 0 “strongly disagree” to 100 “strongly agree”); (iii) Do you consider that you have a problem: With your weight? With your diet? (VAS from 0 “strongly disagree” to 100 “strongly agree”). Lifestyle variables (sleep, physical activity, time using social media, etc.) were measured by asking the number of hours spent before the pandemic and during the first lockdown. For the number of hours of sleep per night, a change (increase or decrease) was defined when there was a difference of at least one hour before and during the first lockdown. For the number of hours of physical activity per week, a change (increase or decrease) was defined when there was a difference of at least 30 min before and during the first lockdown. 

The emotional dimensions tested (anger, sadness, agitation, boredom) were assessed with a VAS from 0 “minimum” to 100 “maximum.” For these variables (visual analogue scales of feelings), the distributions of the variations were divided into three parts, each containing approximately one third of the sample, to classify participants into decrease/no modification/increase.

### 2.4. Statistical Analyses

Categorical data were expressed as number of subjects and associated percentages, and continuous data as mean ± standard deviation or median [25th; 75th percentiles]. To evaluate the evolution of eating and behavior data before and during the first lockdown, results were assigned effect sizes (ES) rather than *p*-values owing to the large sample size, which gave most *p*-values less than 0.001. ES were calculated and interpreted according to Cohen’s recommendations: 0.2 = small effect, 0.5 = medium effect, and 0.8 = large effect [7]. A multiple-correspondence analysis (MCA) followed by a mixed unsupervised classification (*k*-means clustering applied to the partition obtained from an ascending hierarchical classification using Ward’s distance) were also implemented to (i) study the relations between the modalities of the variables and (ii) determine profiles of participants (clusters of individuals sharing closely similar characteristics). For this analysis, the variables were chosen according to univariate results, clinical relevance, and statistical distribution (characteristics always present or always absent were not considered). Only individuals without missing data were included in the MCA, and the sample of excluded subjects was compared to the sample of included subjects, with the chi-squared test or Fisher’s exact test for categorical data, and the Student’s *t*-test or Mann–Whitney test for continuous data.

Statistical analysis was performed using Stata (version 15; StataCorp, College Station, TX, USA) and R 3.5.1 (http://cran.r-project.org/, accessed on 1 March 2022) software. All tests were two-sided, with an alpha level set at 0.05. No missing data were observed for variables related to the primary objective (consumption in the different food categories).

## 3. Results 

### 3.1. Participants

Our study population comprised 671 participants (74% women) with a mean age of 47 ± 13 years. Descriptive characteristics of the participants are shown in Table 1.

The main changes observed during the first lockdown compared to before were an increase in stress level due to COVID-19 (64 [23; 86] vs. 3 [0; 18]; ES: 1.18), a decrease in sleep quality (50 [27; 83] vs. 70 [48; 94]; ES: −0.37), and a deterioration of mood (50 [30; 76] vs. 78 [50; 92]; ES: −0.65). There was also an increase in sedentary behavior (7 [4; 9] vs. 5 [3; 8] hours per day; ES: 0.48) and a decrease in physical activity (2.0 [0.5; 5.0] vs. 3.5 [2.0; 6.0] hours per week; ES: −0.42). An increase in time spent on social media was also observed.

Mood was also modified during lockdown, with more anger (56 [39; 76] vs. 31 [16; 50]; ES: 0.69), more sadness (50 [34; 72] vs. 28 [16; 50]; ES: 0.72), more agitation (50 [25; 66] vs. 43 [20; 50]; ES: 0.29), and more boredom (32 [7; 60] vs. 14 [3; 29]; ES: 0.57). Eating and behavior data for the study population before and during the first lockdown are presented in Table 2 and Figure 1.

### 3.2. Change in Eating Habits during the First Lockdown

The foods whose consumption increased the most between the period before and during the first lockdown were pulses (14.6%), eggs (14.8%), delicatessen (17.9%), alcoholic beverages (25.5%), cocktail snacks (25.6%), and sugary foods (29.2%). The foods whose consumption decreased the most were fish (11.3%), meat (11.8%), and convenience food (11.9%). The changes during lockdown in the different food categories are presented in Figure 2.

### 3.3. Multiple-Correspondence Analysis

For the MCA, 113 out of 671 (16.8%) subjects were removed because of missing data and 558 were retained. These two samples were similar in age, sex, occupational category, weight, and body mass index. The MCA highlighted four groups of subjects, whose characteristics appear in Table 3 and Figure 3 and Figure 4. Group 1 (*n* = 253) included subjects who did not change their eating habits or their number of hours of sleep during the lockdown, who did not have mood changes or variations in their emotions (anger, sadness, agitation, and boredom), and who were more often men than in the other groups. Group 2 (*n* = 155) included nearly all women (90%), whose mood (anger, sadness, agitation, and boredom) deteriorated during the lockdown, and of whom approximately 40% had decreased their sleep time but did not change their eating habits. The proportion of 60–78-year-olds (26%) was higher in this group than in the others. Group 3 (*n* = 91) included “good players,” reducing their consumption of convenience food, starchy foods, snacks, etc., and increasing their consumption of fruits, vegetables, and fish. The subjects in this group also improved their mood (anger, sadness, agitation, and boredom) and increased their sleep time. Group 4 (*n* = 59) included “bad players,” displaying unfavorable changes during the lockdown, with a decreased consumption of fish, pulses, fruits, and vegetables, and an increased consumption of starches, delicatessen, convenience foods, and sugary foods. In this group, we also observed an increase in snacking and a decrease in physical activity.

## 4. Discussion

The present study analyzed the changes in eating habits in France during the first lockdown and patterns of changes in food behavior, lifestyle, and mood. The results of the survey showed that most participants did not change their eating habits during lockdown, but part of the population did significantly modify their eating behaviors. Mood and lifestyle (sleep, physical activity) were also affected.

### 4.1. Change in Eating Habits during the First Lockdown

The study results are in line with other studies showing unfavorable changes during lockdown. The French cohort NutriNet-Santé reported during the first lockdown decreased physical activity (reported by 53% of the participants), increased sedentary time (reported by 63%), increased snacking, decreased consumption of fresh food (especially fruit and fish), and increased consumption of sweets, cookies, and cakes [6]. If we compare the modifications in the consumption of food categories during lockdown (for similar categories) compared to the study of Deschasaux-Tanguy, we see that the consumption of sweet products increased for 29.2% of the participants versus 21.7% (sweets/chocolate), the consumption of alcoholic beverages increased for 25.5% versus 15.4%, and the consumption of fish decreased from 11.3% versus 31.3% (fresh fish). These changes were also confirmed in the CoviPrev survey: less-balanced diet, snacking, and weight gain but also an increase in home-cooked meals. This survey also showed supply difficulties and more attention paid to the food budget: 57% said they found the foods they wanted in stores less often than usual, and 23% paid more attention than usual to their food budget. These developments particularly affected people aged under 40, families with children aged under 16, people with high financial precarity, and people with high levels of anxiety and depression and frequent sleep problems [4]. However, these results conflict with a Spanish study of 945 adults [8], which found that dietary habits improved during lockdown, with a greater adherence to the Mediterranean diet. However, as in our study, confinement had a negative impact on physical-activity levels and sleep quality.

### 4.2. Specific Profiles of Patients during the First Lockdown

Interestingly, among the population, four different patterns were detected according to the modifications undergone during lockdown. Group 1 comprised subjects who did not change their eating habits and lifestyle. Group 2 consisted essentially of women who did not change their eating habits, and whose mood (anger, sadness, agitation, and boredom) deteriorated during the lockdown. Group 3 comprised respondents who had reduced their consumption of convenience food, starchy foods, snacks, etc., and who had increased their consumption of fruits, vegetables, and fish. Group 4, who displayed unfavorable changes during the lockdown, showed a decreased consumption of fish, pulses, fruits, and vegetables and an increased consumption of starches, delicatessen, convenience foods, and sugary foods. In Group 4, we also observed an increase in snacking and a decrease in physical activity. These results are in line with those from the French NutriNet-Santé cohort study, which reported three clusters of participants: Cluster 1 (42.9% of participants) corresponded to those with stable diet-related practices, physical activity, and body weight during the lockdown; Cluster 2 (37.4%) corresponded to participants who exhibited unfavorable nutrition-related changes during the lockdown; and Cluster 3 (19.8%) included participants who reported favorable nutrition-related changes during the lockdown [6]. Group 2 included participants whose different VAS deteriorated during the lockdown. Among the participants in Group 2 (*n* = 155), 115 (74.2%) were angrier during lockdown, 107 (69.0%) were sadder, and 82 (52.9%) were more agitated. Quarantine, which differs from total isolation (despite confusion between the two terms), has negative psychological effects, including post-traumatic stress symptoms, confusion, and anger [9]. Isolation and stress generated by lockdown is accompanied by difficulties in emotional regulation. Lockdown may contribute to the aggravation or development of eating disorders via a reduced capacity for emotional regulation. Flaudias et al. showed a strong relationship between problematic eating behaviors (binge eating and dietary restriction) and stress related to lockdown [10].

People were more frequently confronted with all things related to food and situations associated with eating or that made them think about it. The daily time spent in or near the kitchen was longer than normal. Lockdown increased the time spent on social media and potential exposure to food advertisements. More frequent exposure to food advertising is known to be accompanied by increased cravings, more frequent food intake, and short- and long-term weight gain [11]. Lockdown means storing food in homes, which increases the immediate accessibility and availability of food [12].

In the study by Benamian in Norway [13], female participants were more likely than men to report comfort eating (higher consumption of high-sugar foods and beverages) in relation to COVID-19-related worries and psychological distress. In our study, Group 2 comprised 90.3% women, who suffered emotionally during the lockdown (more anger, sadness, agitation, and boredom) but whose eating habits did not change. In an Italian survey, almost half of the respondents used foods to respond to anxious feelings (48.7%), needed to increase food intake to feel better (55.1%), and had anxious feelings due to current eating habits (57.8%). They also showed that females were more anxious and disposed to eating comfort food than males [14]. This study is consistent with our results showing psychological distress and dietary changes in some patients during lockdown. In another study, the impact of self-quarantine on behaviors was associated with weight gain [15]. Twenty-two percent of the sample gained 5–10 pounds (2.2–4.5 kg). Risk factors for weight gain were identified as inadequate sleep, snacking after dinner, lack of dietary restraint, eating in response to stress, and reduced physical activity [15]. In the present study, no effect of lockdown on weight and body mass index was shown. However, a meta-analysis showed that lockdown affected weight in adult subjects: Body weight was significantly higher (WMD (weighted mean between-group difference) 1.57 (95% CI 1.01 to 2.14)) after the first lockdown period [16]. Here, weight was compared before and during the first lockdown. It would have been of interest to look at weight changes post-lockdown. Another study showed that lockdown increased body weight (+0.62 kg). Some 40% of participants report gaining either 1–4 lbs (0.5–1.8 kg) or >5 lbs (2.3 kg) of body weight [17]. The “weight gainers” had the same profile as the patients in Group 4 of our study, such as frequent consumption of processed food and snacking, and they were less physically active [17].

### 4.3. Limitations

The principal limitation of the study is its retrospective aspect with the reported food consumption by participants. Subjects responding to the questionnaire sometimes responded at a distance from the first lockdown. In this study, only self-reported data were collected, and each respondent could answer anonymously, with no checks possible. However, anonymity may have prevented misreporting.

The cross-sectional nature of this study, which required participants to recall their behaviors prior to and then during lockdown, may have resulted in measurement bias due to memory lapses.

A further limitation relates to the study design (online survey), which may have induced selection bias because participants were volunteers, and those who used the internet more frequently may have been more likely to participate.

The study had more females than males, but it was not possible to control for this imbalance.

Quality control of the COVISTRESS questionnaire was maintained by ensuring only one questionnaire was submitted per IP address, although the same participant might have submitted several questionnaires from different IP addresses.

The changes reported were consistent with those found in other studies on this topic.

## 5. Conclusions

The imposition of lockdown during the COVID-19 pandemic lessened the impact of the virus and limited hospital intensive-care-unit overload. However, psychological concerns and emotional suffering has been regularly reported. In this study, we describe a population clustered into four groups according to their capacities to adapt to the lockdown period regarding their eating habits, food consumption, lifestyle, and mood changes. Among these groups, some subjects seemed to have been specifically impacted by the lockdown. This group should be carefully monitored to prevent deterioration of their quality of life and later deleterious health outcomes. These findings suggest that it is important to define strategies to increase home-based physical activity and encourage populations to eat healthy foods.

## Figures and Tables

**Figure 1 nutrients-14-03739-f001:**
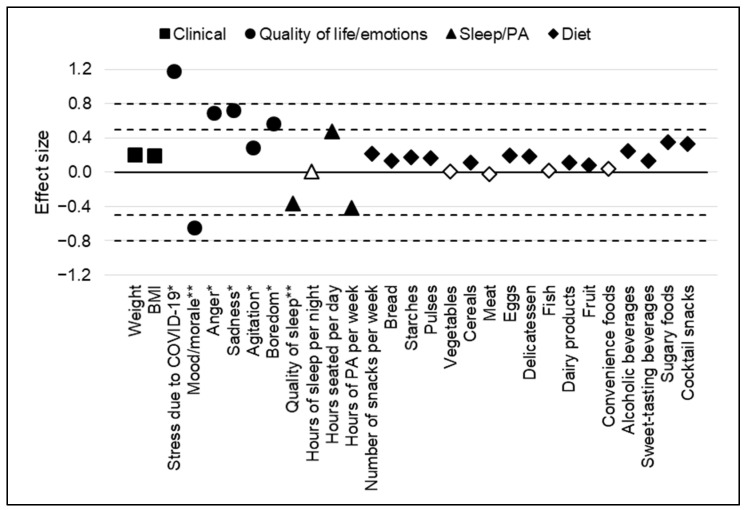
Effect sizes of the variation of each criterion before and during the first lockdown (between 17 March and 10 May 2020). Effect sizes are interpreted according to Cohen’s recommendations: 0.2 = small effect, 0.5 = medium effect, and 0.8 = large effect. Black symbols represent statistically significant effect sizes. White symbols represent statistically non-significant effect sizes. BMI: body mass index; PA: physical activity. * Visual analogue scale from 0 “minimum” to 100 “maximum.” ** Visual analogue scale from 0 “bad” to 100 “excellent”.

**Figure 2 nutrients-14-03739-f002:**
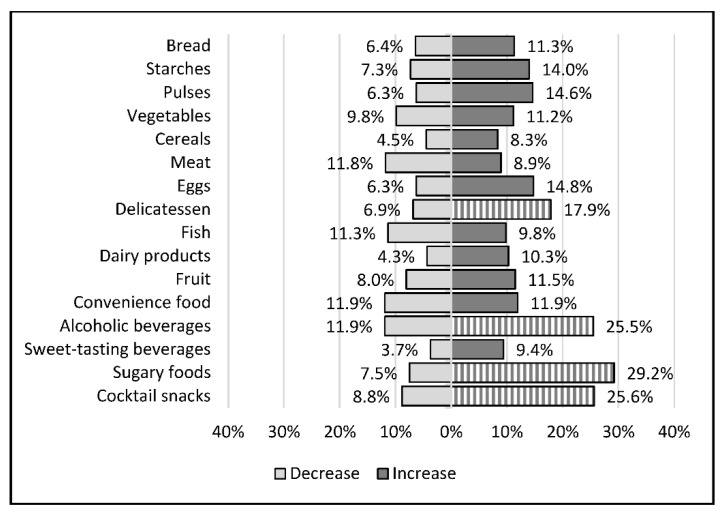
Modifications in the consumption of major food groups during the first lockdown (between 17 March and 10 May 2020). Bars indicate the percentage of participants who reported having increased or decreased their consumption of the food group during lockdown (corresponding number shown on the respective bars). Hatched bars represent percentage above 15%.

**Figure 3 nutrients-14-03739-f003:**
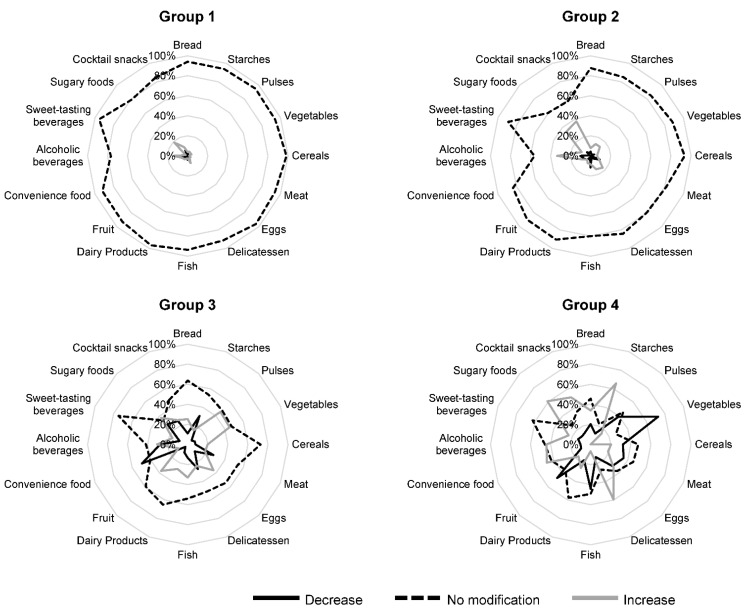
Description of changes in consumption of major food groups before and during the first lockdown, according to the four groups obtained after multiple-correspondence analysis. Black lines represent participants who reported a decrease during the first lockdown compared to before; gray lines represent participants who reported an increase during the first lockdown compared to before, dashed black lines represent participants who reported no modification during the first lockdown compared to before. For food consumption (number of times per week the food is eaten), a change (increase or decrease) was defined when the number was not the same before and during the first lockdown.

**Figure 4 nutrients-14-03739-f004:**
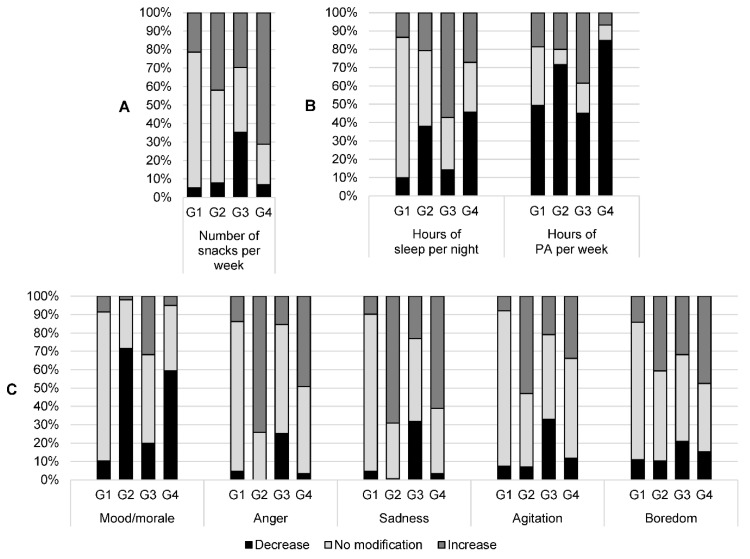
Description of changes in consumption, lifestyle, and mood before and during the first lockdown, according to the four groups obtained after multiple correspondence analysis. (**A**) Number of snacks; (**B**) sleep and physical activity; (**C**) quality of life and mood. Black bars represent participants who reported a decrease during the first lockdown compared to before; light-gray bars represent participants who reported no modification during the first lockdown compared to before; dark-gray bars represent participants who reported an increase during the first lockdown compared to before. For the number of snacks per week, a change (increase or decrease) was defined when the number was not the same before and during the first lockdown. For the number of hours of sleep per night, a change (increase or decrease) was defined when there was a difference of at least one hour before and during the first lockdown. For the number of hours of physical activity per week, a change (increase or decrease) was defined when there was a difference of at least 30 min before and during the first lockdown. For the other variables (visual analogue scales of feelings), the distributions of the variations were divided into three parts, each containing approximately one third of the sample, to classify participants into decrease/no modification/increase. G1 to G4: Group 1 to Group 4; PA: physical activity.

**Table 1 nutrients-14-03739-t001:** Characteristics of the study population (*n* = 671).

	Total (*n* = 671)
Age (years)	47 ± 13
Female sex	494 (74)
Weight (kg)	67 [58; 80]
BMI (kg/m²)	23.5 [20.8; 26.9]
Employment status	
Executive and higher intellectual occupation	230 (34)
Intermediate occupation	130 (19)
Farmer	4 (1)
Artisan, merchant, or entrepreneur	48 (7)
Worker or employee	109 (16)
Student	24 (4)
Job seeker	39 (6)
Retired	69 (10)
MD	18 (3)
Highest educational qualification	
School leaving certificate or under	5 (1)
Youth/vocational training	46 (7)
High school graduation	69 (10)
1st- to 3rd-year university level	256 (38)
5th-year university level: master 2, MDE, or other	135 (20)
Over 5th year of university level: doctorate or other	155 (23)
MD	5 (1)
Type of housing	
House	431 (64)
Flat	207 (31)
MD	33 (5)
Garden	
No	419 (62)
Yes	233 (35)
MD	19 (3)
Household composition	
1 person	128 (19)
2 persons	181 (27)
3 persons	143 (21)
4 persons	106 (16)
5 persons	52 (8)
6 persons	14 (2)
>6 persons	6 (1)
MD	41 (6)

Data are presented as number of subjects (percentages), mean ± standard, or median [25th; 75th percentiles]. BMI: body mass index; MD: missing data; MDE: master’s degree in engineering.

**Table 2 nutrients-14-03739-t002:** Eating and behavior data of the population before and during the first lockdown.

	Pre-Lockdown (*n* = 671)	During the First Lockdown (*n* = 671)
Weight (kg)	67 [58; 80]	67 [58; 80]
BMI (kg/m²)	23.5 [20.8; 26.9]	23.8 [20.8; 27.5]
How do you describe your diet? ^a^	74 [50; 87]	73 [50; 88]
For you, eating is: ^b^		
Essential for living	96 [78; 100]	97 [79; 100]
Useful for maintaining health	97 [82; 100]	97 [83; 100]
A pleasure	96 [79; 100]	95 [76; 100]
A moment of sharing	97 [80; 100]	88 [50; 99]
A constraint	5 [1; 23]	5 [1; 25]
Hours of sleep per night (*n* = 667/667)	7 [7; 8]	7 [6; 8]
Number of snacks per week (*n* = 655/644)	2 [0; 3]	2 [0; 4]
Do you consider that you have a problem ^b^		
With your weight? (*n* = 665/662)	42 [4; 68]	50 [5; 73]
With your diet? (*n* = 661/660)	22 [3; 50]	30 [4; 64]
What is your level of:		
Stress due to COVID-19 ^c^ (*n* = 641/641)	3 [0; 18]	64 [23; 86]
Quality of sleep ^d^ (*n* = 631/638)	70 [48; 94]	50 [27; 83]
Mood/morale ^d^ (*n* = 637/640)	78 [50; 92]	50 [30; 76]
Number of cigarettes per day (*n* = 670/668)	0 [0; 0]	0 [0; 0]
Number of drinks (alcohol) per week (*n* = 670/667)	2.5 [0.0; 2.5]	2.5 [0.0; 2.5]
Number of cannabis uses per week (*n* = 671/670)	0 [0; 0]	0 [0; 0]
Number of hours seated per day (*n* = 647/647)	5 [3; 8]	7 [4; 9]
Number of hours of physical activity per week (*n* = 643/643)	3.5 [2.0; 6.0]	2.0 [0.5; 5.0]
Number of minutes per day using social media		
0	96 (15)	85 (13)
1 to 60	336 (52)	212 (33)
>60	215 (33)	349 (54)
Emotions felt ^c^		
Anger (*n* = 626/638)	31 [16; 50]	56 [39; 76]
Sadness (*n* = 627/634)	28 [16; 50]	50 [34; 72]
Agitation (*n* = 624/632)	43 [20; 50]	50 [25; 66]
Boredom (*n* = 635/642)	14 [3; 29]	32 [7; 60]

Data are presented as number of subjects (percentages) or median [25th; 75th percentiles]. First lockdown: between 17 March and 10 May 2020. BMI: body mass index. ^a^ Visual analogue scale from 0 “not at all balanced” to 100 “very balanced.” ^b^ Visual analogue scale from 0 “strongly disagree” to 100 “strongly agree.” ^c^ Visual analogue scale from 0 “minimum” to 100 “maximum.” ^d^ Visual analogue scale from 0 “bad” to 100 “excellent”.

**Table 3 nutrients-14-03739-t003:** Characteristics of participants in each group obtained after multiple-correspondence analysis.

	G1 (*n* = 253)	G2 (*n* = 155)	G3 (*n* = 91)	G4 (*n* = 59)	*p*
Age (years)					<0.001
18 to 29	34 (13)	9 (6)	22 (24)	8 (14)	
30 to 44	65 (26)	49 (32)	33 (36)	20 (34)	
45 to 59	107 (42)	56 (36)	28 (31)	24 (41)	
60 to 78	47 (19)	41 (26)	8 (9)	7 (12)	
Female sex	159 (63)	140 (90)	67 (74)	49 (83)	<0.001
BMI (kg/m²)					<0.001
<25	159 (63)	117 (75)	47 (52)	29 (49)	
25 to 30	61 (24)	15 (10)	24 (26)	22 (37)	
≥30	33 (13)	23 (15)	20 (22)	8 (14)	
Smoking	42 (17)	18 (12)	18 (20)	12 (20)	0.25
Employment status					0.02
Executive and superior intellectual occupation	100 (40)	55 (35)	24 (26)	20 (34)	
Intermediate occupation	35 (14)	36 (23)	22 (24)	12 (20)	
Artisan, merchant, or entrepreneur	16 (6)	14 (9)	10 (11)	3 (5)	
Worker or employee	55 (22)	13 (8)	17 (19)	10 (17)	
Unemployed ^a^	47 (19)	37 (24)	18 (20)	14 (24)	
Type of housing					<0.001
House	189 (75)	107 (69)	51 (56)	28 (47)	
Flat	64 (25)	48 (31)	40 (44)	31 (53)	

Data are presented as number of subjects (percentages). BMI: body mass index; G1 to G4: Group 1 to Group 4. ^a^ Unemployed: job seeker, retired, student.

## Data Availability

All data generated or analyzed during this study are included in this article. Further enquiries can be directed to the corresponding author.

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
