# Peer review of "Impact of COVID-19 Lockdown on Food Consumption and Behavior in France (COVISTRESS Study)"

_nutrients, 2022, doi:10.3390/nu14183739_

Round 1

Reviewer 1 Report

1- This is a very interesting prospective on  how the pandemic time affected people's behaviors and routine. The background seems to be very generic and I would suggest to re-phrase L 47-66. Also. the description of COVIPREV from line 67 appears disconnected to the previous paragraph. 

2- In line 91-92 the Authors describe a "secondary goal" which is totally unplugged from the previous background paragraph

3- What about the partecipants' privace for their personal details? Please specify it in the text

4- Include a table reference for describing the population at some point from line 95-99

5- Add an appendix to better visualize the questionnaire used and provide a full description of the questions

6- What about the difference between the group 2 and a putative comparison with men with the same characteristic?

7- In relation to the emotional dimension, I would reject the method you used because it is not supported by a validation

Finally- I suggest to read carefully all text in the paper, because the fluency is not sufficient; moreover, the most of the sentences are too long or can be re-phrase. 

Reviewer 2 Report

Dear authors this is a very interesting study on the behavioural and emotional changes observed during the COVID-19 lockdown. However there are several points that need improvement: 

- Introduction: should be extended

- In the procedure section please correct: June 26th, 2020 

- Figure 4: The color description of the bars has been reversed

- Discussion 4.1: Please compare your results within each category with results from other French and international studies.

- Discussion 4.2: Paragraph 3 should be moved to the Introduction

- Discussion 4.2: L. 69-87 should be moved to section 4.1

- Section 4.3: More limitations: cross-sectional study, self-reported data, sample consisted of persons more familiar with the use of internet

- References: Please provide more references from other relevant European studies

- Furthermore, the manuscript would be significantly improved by a thorough english editing.

Round 2

Reviewer 2 Report

Dear authors,

The manuscript is substantially improved.

It just needs an thorough english editing.
